# Change in Self-Esteem Trajectories Among Adolescents and Adults with Intellectual Disabilities

**DOI:** 10.3390/bs15050655

**Published:** 2025-05-12

**Authors:** Eun-Young Park

**Affiliations:** Department of Secondary Special Education, Jeonju University, Jeonju 55069, Republic of Korea; eunyoung@jj.ac.kr

**Keywords:** adolescents and adults with intellectual disabilities, self-esteem, change, longitudinal analysis

## Abstract

Self-esteem significantly influences and shapes an individual’s social behavior and indicates his or her psychological and mental health. In this study, the following two premises are examined: (1) whether the self-esteem of adolescents and adults with intellectual disabilities undergoes changes over time and (2) the demographic variables that influence this change. For this study, a longitudinal analysis spanning 2 years was conducted using response data from 398 participants—as sourced from the Korea Employment Agency for the Disabled—which comprised gender, age, education, and severity of disability as input variables. A potential growth model analysis confirmed the appropriateness of the second measurement change model (significant increase in self-esteem between the first and second measurements, no changes between second and third measurements) that presented the main premises of his study. The findings indicated that the level of self-esteem and its rate of change varied significantly among adolescents and adults with intellectual disabilities, and the variations were primarily associated with gender and severity of disabilities.

## 1. Introduction

As an integral part of the self-concept, self-esteem plays a vital role in human adaptation and mental health, thus affecting an individual’s overall well-being significantly. Most definitions of self-esteem provide an individual’s subjective evaluation of self-worth. [8] ([8]) defined self-esteem as a positive self-concept that involves self-perceptions of capability and value. [11] ([11]) presented self-esteem as an individual’s subjective evaluation of their intrinsic self-worth, regardless of external validation. [41] ([41]) conceptualized self-esteem as an individual’s feeling of being good enough, which entails self-acceptance, self-respect, and self-valuing. [6] ([6]) viewed self-esteem as an evaluative element of self-knowledge and argued that it depends on an individual’s accurate recognition of his or her unique characteristics. However, some individuals cannot evaluate themselves positively with respect to specific value criteria, even if the accuracy of their self-judgment is assessed based on an objective external criterion. In contrast, [42] ([42]) argued that a self-report alone is sufficient, without external objective criteria.

Understanding the dynamics of self-esteem enables effective implementation of interventions aimed at enhancing self-esteem. Despite being stable, self-esteem is not static; it can change in response to various events, experiences, and social environment ([20]; [30]; [45]). While some studies reported that one-third to one-half of adolescents experience low self-esteem during early adolescence ([23]; [25]), it tends to stabilize or increase during middle or late adolescence ([10]). Self-esteem has been reported as very stable during the developmental ages ([7]).

Self-esteem is closely related to an individual’s psychological development. It can vary among individuals in terms of demographic factors such as gender, age, and education. Therefore, analyzing the relationship among the aforementioned factors and self-esteem is important to understand what influences the formation of and changes in self-esteem. Time-invariant variables, such as gender, age, and education, are representative demographic factors related to self-esteem, and gender is reported to show a relatively weak correlation ([4]; [6]). A meta-analysis by [28] ([28]) indicated minimal gender-based effects on self-esteem, and men exhibited self-esteem slightly more positively. With respect to gender, girls tend to have lower self-esteem than boys, and this difference appears to widen in late adolescence ([13]). With respect to age, self-esteem is high in childhood, decreases in adolescence, and gradually increases throughout adulthood ([40]). With respect to strong correlation of education level with self-esteem than income ([46]) individuals with higher educational levels exhibit a higher self-esteem trajectory ([36]).

Despite its importance as a psychological construct and association with mental and physical health, little is known about self-esteem among adults with intellectual disabilities ([29]). American Intellectual and Developmental Disabilities characterizes intellectual disability by significant limitations in intellectual functioning and adaptive behavior, which is expressed in conceptual, social, and practical adaptive skills ([43]). Most studies on self-esteem in people with intellectual disabilities have focused on analyzing the level of self-esteem at a specific point in time or the correlations among related factors. While these studies have provided basic data for understanding the characteristics of self-esteem in people with intellectual disabilities, they could not effectively capture changes with respect to time or individuals. [18] ([18]) analyzed individuals’ differences in terms of self-concept, self-esteem, and psychopathology symptoms in two groups, with and without intellectual disabilities. They reported that the group with intellectual disabilities had significantly lower self-concept and self-esteem and exhibited most psychopathology symptoms. Furthermore, gender-based differences were insignificant in both groups. This is a cross-sectional study with data obtained at the same point in time; thus, it cannot identify changes in self-concept or self-esteem over time. Only 42 out of the 170 participants were intellectually disabled, limiting the results’ generalizability. A systematic literature review revealed that school-aged youth with intellectual disabilities had lower cognitive, academic, and self-esteem scores than typically developing adolescents, and recognized few significant relationships of self-concept with academic achievement and gender ([31]). This study explored self-concept and self-esteem in adolescence; however, it may have missed changes or developmental perspectives over time due to the lack of a longitudinal approach. Studies on self-esteem targeting general public are often cross-sectional and longitudinal, and finding research on the developmental trajectory of self-esteem among people with intellectual disabilities remains difficult. Previous cross-sectional studies on self-esteem in people with intellectual disabilities suggest limitations in interpreting causal relationships and highlight the need for longitudinal studies to identify changes over time, because self-concept and self-esteem are developmental in nature ([9]; [44]).

Self-esteem is a key element in human psychological development that affects mental health, social relationships, and quality of life ([34]). In a study of 619 people with intellectual disabilities, happiness and a positive outlook on life were significantly associated with positive self-perception and evaluations from others, high self-esteem, and a sense of autonomy ([3]). [2] ([2]) reported that the higher the self-determination of students with disabilities, the higher their physical and psychological health, self-esteem, and overall satisfaction in life ([43]). In general, self-esteem tends to be stable or increase during the transition from childhood to adolescence and adulthood ([15]). However, such tendency of general population with typical development may not appear in people with intellectual disabilities. In the middle adolescence period, from the age 15 onwards, self-concept becomes increasingly sophisticated and stable ([24]; [33]), and social factors, such as peer relationships and school experiences, significantly influence self-esteem ([39]). In the transitional period of adolescence, people with intellectual disabilities may experience difficulties in social integration, identity formation, and emotional regulation ([12]). Hence, this study examines change aspects of self-esteem in adolescents and adults with intellectual disabilities and aims to understand their psychological characteristics in detail.

Despite realizing the importance of longitudinal studies, research on long-term changes in the self-esteem of adolescents and adults with intellectual disabilities remains lacking because of insufficient highly representative panel data, making systematic verification of these conflicting claims difficult. In a previous study targeting adults with intellectual disabilities, 25 papers were analyzed, and mixed evidence was obtained regarding their self-esteem levels. The study suggested that factors such as engagement in life were related to higher self-esteem, and low self-esteem were associated with depression ([29]). In particular, understanding the changeability in self-esteem of adolescents and adults with intellectual disabilities remains difficult. To address this knowledge gap, the current study used panel data to determine the patterns of change in the self-esteem of adolescents and adults with intellectual disabilities and the demographic variables influencing such change.

Thus, this study aims to answer the following research questions: (1) Does the self-esteem of adolescents and adults with intellectual disabilities change over time? (2) Which demographic variables influence changes in self-esteem among adolescents and adults with intellectual disabilities?

## 2. Materials and Methods

### 2.1. Study Design

In this study, a longitudinal design was employed.

### 2.2. Data

In this study, the first and second waves of the 8th Panel Survey on Employment for the Disabled data provided by the Korea Employment Agency for the Disabled were used to determine changes in self-esteem among adolescents and adults with intellectual disabilities. The second-wave data were obtained from a 2018 survey because of the accumulation of structural problems of the longitudinal design in the first-wave survey (the occurrence of withdrawal due to unavoidable reasons such as death, loss of the youth population due to a natural increase in age, aging of the disabled, etc.) hindered the representation of people with disabilities in the survey results. This survey included the first, second, and third measurements in 2018, 2019, and 2020, respectively. The inclusion criteria comprised individuals with disabilities who were aged 15–64 years and were registered under the Welfare of Persons with Disabilities Act of the Republic of Korea. Because the Republic of Korea has adopted a disability registration system in which a doctor’s diagnosis of a disability is required for registration, participation was restricted to individuals who had received a physician-diagnosed intellectual disability.

A two-stage sampling method was employed to control the number of extracted areas and samples according to disability type, severity, and age. Disability samples were selected using a one-stage colony sampling method, followed by stratification based on disability type, severity, and age. The stratification was performed to meet the margin level of the target error. Of 4577 panel participants, 398 had intellectual disabilities. The target group included adolescents and adults with intellectual disabilities who lived in the community, while none lived in facilities. In the second measurement, attribution occurred and 374 people responded. In the third measurement, no further attribution occurred, and the same 374 people responded. The appropriate sample size for the latent growth model may vary depending on the complexity of the model and the number of free parameters; however samples of at least 100 are recommended ([21]).

The survey was conducted using a tablet PC-assisted personal interviewing system. After visiting the panel, the interviewer used a tablet PC to ask questions from the respondents according to a structured questionnaire and saved their responses on the computer to collect data. Before conducting the survey, a preliminary preference survey was conducted through telephone in which the respondents’ information, address, and contact information were confirmed. Then, the main survey was initiated, and the visit date was set in advance for each participant. The survey followed a data verification procedure in the following five stages: (1) by the tablet PC-assisted personal interviewing system, (2) self-verification (3) through supervisor review, (4) by the independent telephone verification team, and (5) through logical errors and comparative data. Discovery of an error during the verification process required reconfirmation with the panel, another round of survey was conducted via telephone.

### 2.3. Measurement

In this study, the [41] ([41]) Self-Esteem Scale was used to measure self-esteem. After initially developing the scale in 1965, Rosenberg revised and supplemented it in 1989, allowing researchers worldwide to use it freely, without copyright. This self-reported scale measures individuals’ attitudes toward themselves and comprises 5 positive and 5 negative self-esteem questions. Responses are rated on a five-point Likert-type scale ranging from “not at all” (1) to “very much” (4). A higher overall score indicates higher self-esteem, and negative self-esteem questions, 3, 5, 8, 9, and 10, are scored using reverse calculations. By analyzing the second-wave measurement data of the panel, self-esteem levels of adolescents and adults with intellectual disabilities were confirmed and ascertained. [37] ([37]) found an appropriate item fit, rating scale, and separate index (reliability) for measuring self-esteem through Rasch analysis and confirmed the scale as a useful tool for the self-esteem of adolescents and adults with intellectual disabilities. Rosenberg self-esteem scale structure composed of a four-point Likert scale for elementary school students with mild intellectual disabilities and confirmed the validity of the bifactor model ([44]). The scale demonstrated excellent reliability with Cronbach’s α values of 0.976 (0.973–0.980), 0.983 (0.980–0.985), and 0.984 (0.981–0.986) for the first, second, and third measurements, respectively.

The obtained demographic information included gender (male was coded as 1 and female as 2), age (15–29 years, 30–39 years, 40–49 years, 50–59 years, and above 60 years), educational level (below middle school graduate, high school graduate, and above college graduate), severity of disability (Levels 1, 2, and 3), presence of comorbid disability (yes, no), residential area (big city, small city, medium-sized city, and rural), and marital status (unmarried, married, divorced, and bereaved). For analyzing the conditional latent growth model, age was the input as a continuous variable.

### 2.4. Statistical Analysis

Descriptive statistics (means, standard deviations, and correlations) were used to analyze the survey data of individuals with intellectual disabilities using SPSS 26.0 and AMOS 26.0. The latent growth model examined changes in self-esteem of adolescents and adults with intellectual disabilities over time and investigated the variables affecting this change. The basic linear growth model was first tested using panel survey data on self-esteem from the three measurements in which the slope of the basic linear model changed; therefore, the constant term was fixed to 1. The slope assumed a linear pattern of change and was designated as 1, 2, and 3 for first, second, and third measurements, respectively. The second measurement change model assumed a change in the slope between the first and second measurements; however, there were no changes between second and third measurements. Therefore, the slope of the first measurement was designated as 0, and those of second and third measurements were designated as 1. The third measurement change model assumed no change between first and second measurements and a change in slope in the third measurement. Therefore, the slopes of the first and second years were designated as 0, and that of the third measurement was designated as 1. The second and third measurement change model were considered as a nonlinear model.

An unconditional latent growth model was used to identify a suitable model to explain the change in the self-esteem of adolescents and adults with intellectual disabilities among first, second, and third measurement change models. Conditional latent growth model was used to identify the effects of gender, age, and education level on the initial (intercept) and subsequent change in self-esteem over time (slope) among adolescents and adults with intellectual disabilities.

In this study, the comparative fit index (CFI), normed fit index (NFI), and root mean square error of approximation (RMSEA) were used as fit indices. The fit indices of values more than 0.90, were considered a good fit. For RMSEA, a value more than 0.10 was considered a poor fit and a value from 0.05 to 0.08 was considered acceptable ([26]). After examining the fit of the model for changes in self-esteem among adolescents and adults with intellectual disabilities, the relationship between the change models and demographic variables was verified.

The full information maximum likelihood (FIML) method was employed to address the missing values, as it estimates the characteristics of the population by considering the pattern of missing values appropriately even under the assumptions of completely randomized longitudinal study ([14]). In this study, 24 data points from the second and third waves were missing.

## 3. Results

### 3.1. Descriptive Statistics

The study included 253 men (63.6%) and 145 women (36.4%). The majority of participants were aged 15–29 years 203 (51.0%); 211 (53.0%) were high school graduates; and 34 (8.5%) had concomitant disabilities. Most participants (338; 84.9%) were unmarried and lived in large cities (168; 42.2%) (Table 1). The results showed significant differences based on the participants’ gender and residential area in the first measurement and based on the severity of disabilities and residential area in the second and third measurements.

Correlation analysis revealed significant relationships between gender and the first measurement of self-esteem, age and education level, first and second self-esteem measurements, and second and third self-esteem measurements (*p* < 0.05).

### 3.2. Analysis of Change in Self-Esteem and General Characteristics

The model’s appropriateness for analyzing changes in self-esteem was verified using confirmatory factor analysis and justified based on fit indices (Figure 1). Table 2 shows the results regarding the goodness-of-fit of the first, second, and third measurement change model. Upon comprehensive examination of the values of *χ*^2^, RMSEA, NFI, and CFI, the measurement change model provided the best fit for self-esteem.

Table 3 shows the changes in self-esteem based on the second measurement change model. This means that a significant increase in self-esteem occurred between the first and second measurements; however, there were no changes between the second and third measurements. The initial self-esteem value was 29.864 (*p* < 0.001), with a variance of 89.873 (*p* < 0.001), and the rate of change was 2.777 (*p* < 0.05), with a variance of 237.905 (*p* < 0.001). The significant initial values for mean and variance indicated individual differences in self-esteem. The mean of the significant rate of change indicated significant change in self-esteem over the 2 years, and the significant variance indicated that the trajectory of change in self-esteem over the 2 years has shown significant differences among adolescents and adults with intellectual disabilities. The significance of the covariate reflected the relationship between the initial value and the rate of change.

### 3.3. Relationship Between Self-Esteem and Demographic Characteristics

The results of the relationship between changes in self-esteem and demographic characteristics are provided in Figure 2. The appropriateness of the model was verified, as the value of *χ*^2^ was 6.859 (*p* = 0.334), and the values of NFI, CFI, and RMSEA were 0.960, 0.997, and 0.013, respectively. The results showed that gender was associated with the initial self-esteem level (*B* = −3.907, *p* < 0.05) and changes in self-esteem (*B* = 4.696, *p* < 0.05). Gender was negatively correlated with initial self-esteem, indicating that females had lower self-esteem than males at the first measurement, and was positively correlated with the rate of change in self-esteem, indicating that self-esteem of females increased more quickly than males. Severity of disability was associated with changes in self-esteem (*B* = −3.630, *p* < 0.05).

## 4. Discussion

Because a longitudinal analysis is crucial for developing interventions to promote healthy self-esteem by identifying the major factors that improve or worsen self-esteem, previous research included longitudinal studies investigating changes in self-esteem and describing its patterns in general population ([5]; [15]; [19]). However, such studies in people with intellectual disabilities are lacking. This study is the first to apply growth curve modeling to examine whether the self-esteem among adolescents and adults with intellectual disabilities changes over time using a longitudinal analysis and whether demographic variables affect this change. The rationale of analysis starting from individuals of 15 years of age is that in this developmental period, self-concept is stabilized and social influence is strengthened ([33]).

The findings indicate that the self-esteem of adolescents and adults with intellectual disabilities change over time in a nonlinear fashion. In particular, self-esteem underwent significant change between first and second measurements but remained stable between second and third measurements. The change between first and second measurements suggest that individuals’ self-esteem increased significantly as they initially responded to the intervention or changes in daily life. Stabilization after the second measurement indicates stabilized self-esteem. In addition, initially low self-esteem may indicate that no further significant change occurred after it improved to a certain level. Gradual psychological changes in terms of self-esteem show a flattened growth curve at a certain level. A longitudinal study examining youth and adults reported that changes in self-esteem are nonlinear, and they slow down with increasing age ([15]). Another study revealed that self-esteem exhibits linear growth between 16 and 32 years, peaks thereafter, and shows no significant growth between 32 and 42 years ([27]). [5] ([5]) suggested a curvilinear relationship between age and self-esteem, indicating the dynamic nature of self-esteem during adolescence. [27] ([27]) reported a quadratic self-esteem model using a linear model suggesting nonlinear change in self-esteem. The average initial values of self-esteem of adolescents and adults with intellectual disabilities was 29.864 and 87.873, respectively, and these significant values demonstrated significant individual differences. The average rate of change was 2.777, showing that the self-esteem of adolescents and adults with intellectual disabilities tended to increase over time. Furthermore, the significant variation in the rate of change (237.905) indicated significant individual differences in terms of the rate of change in the self-esteem of adolescents and adults with intellectual disabilities. The correlation between the initial value and the rate of change was negative and significant at −62.459, indicating that the higher the level of self-esteem in the first year, the lower the increase in self-esteem. The results of this study are consistent with those of previous studies targeting the general public and reporting that self-esteem decreases during childhood and adolescence owing to confusion in confidence and identity caused by changes in various physical and social cognitive characteristics [16] ([16]). while it stabilizes or increases during adulthood ([19]). Relatively short-term longitudinal analysis studies have reported that self-esteem gradually increases during middle age ([32]; [47]).

Demographic factors such as gender, age, and education, along with severity of the disability, affect individuals’ self-esteem ([1]; [22]). These factors are considered appropriate for examining change patterns in the latent growth model due to their time-invariant characteristics. The negative relationship between gender and the initial self-esteem level showed that women have lower self-esteem than men. The positive relationship between gender and the rate of change in self-esteem showing that increase in self-esteem was faster among women with intellectual disabilities than their male counterparts. The results of previous longitudinal and cross-sectional studies on the effects of gender on self-esteem have yielded conflicting results. Some longitudinal studies have reported small gender differences in adolescence ([38]) and young adulthood ([36]; [40]), while others have reported no significant gender effects ([15]). Some cross-sectional studies reported gender-based differences in self-esteem ([17]), while others reported no such differences ([18]) The nonsignificant age-related results of this study differed from those of previous studies, as [5] ([5]) demonstrated the significant effect of age on self-esteem in a growth curve analysis. The differences may be due to the differences in the participants’ age range, which was 15–65 years for this study and 11–16 years for the study by [5] ([5]). A meta-analysis of longitudinal studies reported that participants with higher educational levels had higher self-esteem at all ages ([33]). The trajectory of self-esteem among individuals with higher education levels consistently remained exhibited a curvilinear trend for the groups with higher and lower educational levels ([35]). In this study, no difference in self-esteem was observed in terms of severity of disability, and self-esteem was found to decrease with the increasing degree of disability. Such negative relationship between the degree of disability and self-esteem has also been reported in a cross-sectional study ([18]). The changes in self-esteem based on gender and the severity of the disability may be due to the initial low self-esteem value in the first measurement. In cases of initially low self-esteem, individuals may react sensitively to psychological intervention, social support, and environmental changes, exhibiting significant changes.

Although self-esteem is recognized as a significant psychological construct with implications for mental and physical health in the general population, only a limited understanding has been acquired of its manifestations in adolescents and adults with intellectual disabilities ([29]). The results of this study showed that the self-esteem of adolescents and adults with intellectual disabilities increased over time, and gender was identified as a key factor affecting the initial level and changes in self-esteem.

### 4.1. Limitations and Directions for Future Research

This study has several limitations. First, since this study utilized panel data, which is secondary data, it does not sufficiently reflect the various demographic and environmental variables that may affect changes in self-esteem. For example, family support, peer relationships, quality of special education services, and level of community integration are major factors that may affect self-esteem; however, they were not included in these data. Thus, future studies need to collect primary data that includes a wider range of variables and analyze other factors that affect self-esteem. Second, the sample size in this study was insufficient to divide the intellectually disabled individuals according to their age groups for better analysis; hence, life cycle–related change trajectory in self-esteem could not be clearly identified. In future studies, a large-scale longitudinal sample that sufficiently includes the intellectually disabled by age group should be secured to enable a comparative analysis of the developmental trajectory of self-esteem throughout the life cycle. Third limitation is that individual differences in terms of rate or pattern of change in self-esteem among individual participants could not be analyzed precisely. Therefore, future studies should also utilize analytical techniques, such as latent class growth modeling, to capture not only intraindividual but also interindividual changes.

### 4.2. Implications for Practice

The results of this study provide the following implications for practical implementation supporting intellectually disabled adolescents and adults:

First, self-esteem should be recognized as a dynamic psychological aspect that requires continuous psychological and emotional support. Hence, systematic interventions to improve self-esteem are required in schools, welfare organizations, and adult transition support centers. In particular, the results confirming the existence of intraindividual differences suggest the need for individually tailored training programs to enhance self-esteem. Second, this study found that gender and the severity of disability affected changes in self-esteem, and the rate of change was particularly high in women with low initial self-esteem levels, suggesting the need for customized self-esteem improvement programs tailored to individual characteristics, especially gender and severity of disability. In particular, programs such as peer support groups, emotional expression training, and expanded opportunities for self-expression targeting female adolescents and adults with intellectual disabilities can have a positive effect. Third, since self-esteem is closely related to an individual’s social integration, identity formation, and emotional stability, interventions to improve self-esteem can go beyond simple psychological support and enhance overall quality of life. Therefore, systematically designing opportunities for positive interaction, accumulation of small-scale success experiences, and experiences of social recognition remains essential. Fourth, in this study, self-esteem significantly increased between the first and second measurements, and there was no change between the second and third measurements, showing that self-esteem interventions can work more effectively in the early stages.

## 5. Conclusions

This longitudinal study differs from previous short-term or cross-sectional studies that focus on the change in self-esteem of adolescents and adults with intellectual disabilities. The results confirm that the self-esteem development trajectories of adolescents and adults with intellectual disabilities differ among individuals. The results further showed a significant change between first and second measurements, and gender and severity of disability in adolescents and adults with intellectual disabilities might have had an effect on changes in their self-esteem. These results contribute to a deeper understanding of the psychological development of individuals with intellectual disabilities by presenting meaningful empirical evidence that self-esteem remains dynamic throughout life. In particular, in the case of adolescents and adults with intellectual disabilities with low self-esteem, a high rate of change in women’s self-esteem indicates that providing programs to improve self-esteem can be helpful.

## Figures and Tables

**Figure 1 behavsci-15-00655-f001:**
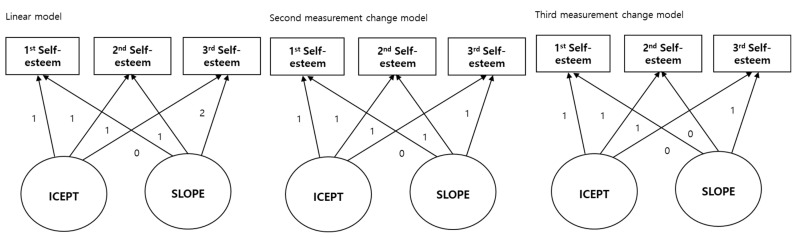
Three hypothesized latent growth models.

**Figure 2 behavsci-15-00655-f002:**
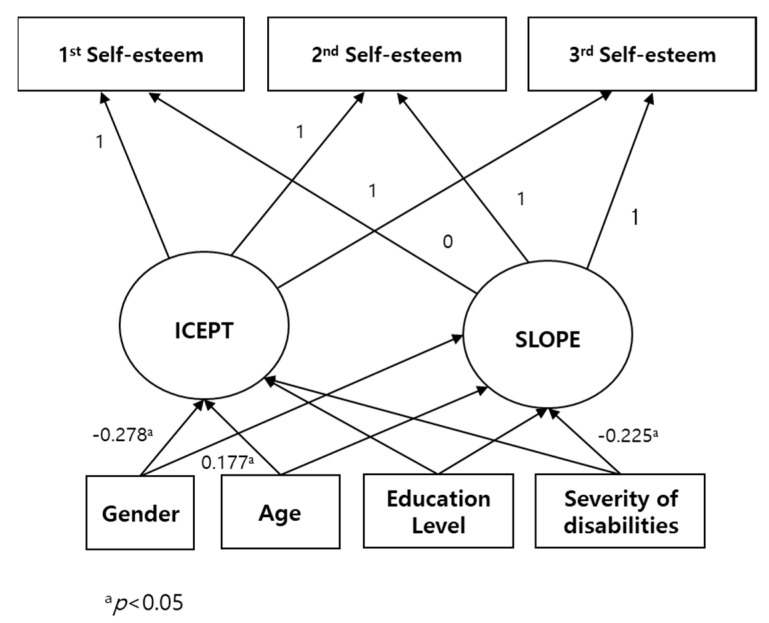
The relationship between change in self-esteem and demographic characteristics.

**Table 1 behavsci-15-00655-t001:** General characteristics of individuals with intellectual disabilities.

Categories	n	%	χ^2^	1st Self-Esteem			2nd Self-Esteem			3rd Self-Esteem		
				M	SD	t/F	M	SD	t/F	M	SD	t/F
Gender												
Male	253	63.6	29.307 **	31.10	19.522	2.052 *	31.46	19.14	−0.910	33.22	20.59	0.417
Female	145	36.4		27.39	12.714		33.43	21.79		32.29	20.84	
Age												
15~29	203	51.0	304.965 **	30.60	17.638	0.618	32.67	19.68	0.496	33.42	20.37	0.362
30~39	105	26.4		29.52	18.096		31.82	19.68		32.44	20.40	
40~49	49	12.3		27.41	13.873		33.92	23.87		34.31	23.49	
50~59	36	9.0		29.97	19.706		29.66	20.13		30.12	20.74	
60~64	5	1.3		21.60	5.128		32.67	19.68		26.60	2.88	
Severity of disabilities ^+^												
Level 1	96	24.1	186.156 **	29.81	18.79	0.243	36.72	26.30	3.299 *	39.10	27.07	5.536 **
Level 2	137	34.4		28.96	16.58		31.18	18.95		31.27	18.63	
Level 3	165	41.5		30.37	17.38		30.23	16.04		30.62	17.09	
Education level												
Below middle school graduate	168	42.2	153.050 **	29.13	17.451	0.442	32.51	21.65	0.918	33.08	22.04	0.975
High school graduate	211	53.0		29.96	17.187		31.82	19.18		32.80	19.89	
Above college graduate	19	4.8		32.95	20.315		33.39	17.44		32.00	16.96	
Comorbid disability												
Yes	34	8.5	273.618 **	26.91	11.753	−0.993	36.12	24.24	1.175	37.56	24.99	1.386
No	364	91.5		30.02	17.855		31.81	19.71		32.41	20.15	
Residential area												
Big city	168	42.2	29.553 **	35.44	23.175	17.090 **	36.96	24.28	8.168 **	32.19	20.15	4.468 *
Small and medium-sized city	83	20.9		24.37	3.724		27.11	13.25		35.01	22.45	
Rural	147	36.9		26.29	11.749		29.76	17.08		26.79	11.25	
Marital status												
Unmarried	338	84.9	769.136 **	30.38	18.430	1.254	32.65	20.59	0.564	33.51	21.40	0.487
Married	41	10.3		27.51	10.904		29.95	17.39		28.85	14.90	
Divorce	13	3.3		22.77	3.940		31.69	22.06		32.54	20.22	
Bereavement	6	1.5		24.50	2.74		23.83	3.87		25.40	2.07	

Note. ^+^ Level 1: IQ below 35, severe adaptation difficulties in daily life and lifelong care required; Level 2: IQ 35~49, tasks can be performed with continuous supervision or special training; Level 3: IQ 50~70, social and vocational rehabilitation possible through education; * *p* < 0.05; ** *p* < 0.01.

**Table 2 behavsci-15-00655-t002:** Model appropriateness for self-esteem.

Model	*χ^2^*	*Df*	*NFI*	*CFI*	RMSEA (LO90~HI90)
Linear model	21.46	2	0.836	0.844	0.155 (1.00~1.218)
Second measurement change model	0.725	2	0.994	1.000	0.000 (0.00~0.074)
Third measurement change model	61.403	2	0.524	0.525	0.221 (0.175~0.271)

Note: NFI = normed fit index; CFI = comparative fit index; RMSEA = root mean square error of approximation.

**Table 3 behavsci-15-00655-t003:** Estimates of the second measurement model for change in self-esteem.

Selected Model	Category	Mean	Variance	Covariance
Second measurement change model	Intercept	29.864 ***	87.873 ***	−62.459 ***
	Slope	2.777 *	237.905 ***	

Note: * *p* < 0.05; *** *p* < 0.001.

## Data Availability

https://edi.kead.or.kr/BoardType17.do?bid=18&mid=37 (accessed on 10 November 2024). To download a data file, connect to the site and request data.

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
