# Peer review of "Change in Self-Esteem Trajectories Among Adolescents and Adults with Intellectual Disabilities"

_behavsci, 2025, doi:10.3390/bs15050655_

Round 1

Reviewer 1 Report

Comments and Suggestions for Authors

This is a very interesting research topic. Several issues should be adressed by the authors.

1) What this study will add to previous research about longitudinal changes of self-esteem among people with intellectual disabilities ? The authors must give more information about findings from previous studies (strength and weakness) and explain why a trajectory study is necessary.

2) This not clear why the authors are focusing on adolescents and adults. Chenges in self-esteem may differ during these developmental period. Moreover, the sample of adolescents seems to be limited in number (those age between 15 to 17 years old). The authors should rather focus on adults.

3) Please, provide information of the severity of the intellectual disability. This is an important information that is lacking. This variable should also be included as predictors of the intercept and slope factors in the analysis. It is frequently reported in the litterature that levels of self-esteem differ as function of the severity of the intecctual disability.

4) Information about the latent growth curve model are unsufficient. How, the model was configured should me more clearly explain. It is unclear why the slope have been coded 0, 1, 1 and not 0, 1, 2 if data were collected at 1 year of interval. Several times in the manuscript the authors mention "over two years" whereas the abstract mention that it is a three year study.

5) For the latent growth curve model (LGM), please use a terminology frequently used in the litterature: unconditional LGM and conditional LGM with predictor of the intercept and slope factors.

6) Table 1 present the mean and SD of self-esteem for the first year only. Please, provide the information for all measurement points.

7) In table 2, what does "Wave 2 variation model" and "Wave 3 variation model" mean? 

8) In table 3, what does "Wave 2 variation model"  mean ?

9) Please, explain how sex was coded (e.g., boys = 0 and girls = 1) ? This will be helpful to undrestand the results on Table 4.

10) This not clear why "age" has not been used as a continuous variable. This make more sens than using arbitrary age category. 

11) The discussion must integrate more litterature about changes in self-esteem among people with intellectual disabilities.

Author Response

Overall: This is a very interesting research topic. Several issues should be addressed by the authors.

Response: I sincerely thank you for taking the time to carefully review this manuscript. I read your valuable opinions, carefully reviewed the contents of the manuscript accordingly, and revised them in good faith. I would like to say that your advice has been very helpful in improving the completeness of this study, and I would like to express my sincere gratitude once again. The revised text in the manuscript has been highlighted in blue font. The responses to all comments have been prepared and attached herewith/given below.

Comment #1: What this study will add to previous research about longitudinal changes of self-esteem among people with intellectual disabilities? The authors must give more information about findings from previous studies (strength and weakness) and explain why a trajectory study is necessary.

Response #1: Regarding to this comment, rationale of this longitudinal study has been expanded based on findings form previous studies (page 2, line 25 ~ line 46).

Comment #2: This not clear why the authors are focusing on adolescents and adults. Changes in self-esteem may differ during these developmental periods. Moreover, the sample of adolescents seems to be limited in number (those age between 15 to 17 years old). The authors should rather focus on adults.

Response #3: The reason why this study is focusing on adolescents and adults with intellectual disabilities have been added (page 3, line 12 ~ line 23).

Comment #3: Please, provide information of the severity of the intellectual disability. This is an important information that is lacking. This variable should also be included as predictors of the intercept and slope factors in the analysis. It is frequently reported in the literature that levels of self-esteem differ as function of the severity of the intellectual disability.

Response #3: Information on the degree of intellectual disability is presented by inserting it into the table. Although information on disability grade is provided in the panel data, the disability grade system for intellectual disability has been abolished in South Korea and is divided into severe and mild. All individuals with intellectual disabilities are severe (Table 1, Figure 1).

Comment #4 & 5: Information about the latent growth curve model is insufficient. How, the model was configured should me more clearly explain. It is unclear why the slope has been coded 0, 1, 1 and not 0, 1, 2 if data were collected at 1 year of interval. Several times in the manuscript the authors mention "over two years" whereas the abstract mention that it is a three-year study. For the latent growth curve model (LGM), please use a terminology frequently used in the literature: unconditional LGM and conditional LGM with predictor of the intercept and slope factors.

Response #4 & 5: Related content has been supplemented and expanded, and terms have been redefined and reused (page 5, line 20 ~ line 31; Throughout manuscript).

Comment #6: Table 1 present the mean and SD of self-esteem for the first year only. Please, provide the information for all measurement points.

Response #6: All three measurement values ​​are presented for each variable (Table 1).

Comment # 7 & 8: In table 2, what does "Wave 2 variation model" and "Wave 3 variation model" mean? In table 3, what does "Wave 2 variation model" mean?

Response # 7 & 8: The terminology was changed to Second measurement change model and Third measurement change model, and presented in detail in the model section of statistical analysis (page 5, line 20 ~ line 31; Throughout manuscript).

Comment #9: Please, explain how sex was coded (e.g., boys = 0 and girls = 1)? This will be helpful to understand the results on Table 4.

Response #9: The information about coding have been added (page 5, line 7 ~ line 8).

Comment #10: This not clear why "age" has not been used as a continuous variable. This makes more sense than using arbitrary age category. 

Response #10: I reanalyzed the data using age as a continuous variable and presented the results in a revised form (page 5, line 12 ~ line 13).

Comment #11: The discussion must integrate more literature about changes in self-esteem among people with intellectual disabilities.

Response #11: The content of the discussion was revised overall to reflect the opinions, and information on related literature was also added (page 9, line 36 ~ line 39; page 10, line 10 ~ line 17).

Reviewer 2 Report

Comments and Suggestions for Authors

Thank you for the opportunity to review this article. My comments are shown below and I hope the authors find them helpful to improve the manuscript.

Brief Synopsis: This manuscript presents findings from a longitudinal analysis examining the change of self-reported levels of self-esteem in a sample of individuals with intellectual disabilities. This study uses data from a Panel Survey through the Korea Employment Agency for the Disabled. Findings suggest self-esteem in adolescents and adults with intellectual disabilities demonstrate a non-linear change over time, and females reported lower rates of self-esteem yet faster rates of changes compare to males with intellectual disabilities. The discussion focuses on the comparison of findings to previous, inconsistent results, and suggests future direction needed based on the identified limitations of the current study. 

Abstract: Must be updated and clarified based on the comments below. Make sure to spell out AAWID before using the abbreviation (avoid using abbreviation in the abstract).

Introduction: The introduction does a good job on explaining the differential definitions of self-esteem and theories of the trajectory of self-esteem across the lifespan. Consider expanding the introduction to address specific findings for individuals with ID or other related factors (such as self-determination, quality of life, etc.).

  1. It appears there is a lack of knowledge of the trajectory of self-esteem in individuals with ID but it may be helpful to briefly describe the current state of what is known about self-esteem generally in ID and its association with larger constructs such as well-being or quality of life.

Methods:                                                             

The authors methodology and use of latent growth curve modeling and descriptions of fit indices is clear and well thought out. However, there are some minor inconsistencies throughout this section that should be addressed.

  1. There is some inconsistency in the design and data used in this study. In the “Study Design” subsection of the Materials and Methods section, The author states the study employed “a cross-sectional survey design”. In the “Data” subsection, the authors state they used data from wave 2 of the survey. However, based on the details provided further into the methods and the results, it does not appear this study is a cross-sectional survey design, and waves 1 and 3 were also reported.
    1. More detail is needed to describe each wave of data used, the time in between data collection, and any attrition between the data collection periods.
  2. What is the rationale for the age ranges and age groupings selected?
  3. Self-Esteem Scale:
    1. Given the challenges they may arise from individuals with intellectual disabilities ability to self-report, were their any tests of comprehension or adaptations made to ensure the individuals were able to effectively and accurately report on their own experiences?
    2. It may be helpful to report psychometrics related to individuals with intellectual disabilities for this scale
    3. It appears this scale originally uses a 4-point Likert scale, was this an adaptation made by the author?
  4. The severity level of Intellectual disability would be beneficial to report with this sample and determine if level of ID impacts in changes of self-esteem over time.
  5. Statistical Analysis:
    1. “the comparative fit index (CFI), normed fit index (NFI), and root mean square error of approximation (RMSEA) were used as fitness indices. If the fit indices are >.90, the model is considered to be a good fit. In the case of RMSEA, a good fit is when the value <.10 is considered a bad fit”. These sentences are unclear and seem to be contradictory. RMSEA’s values > .10 indicates a poor fit, while 0.05 and 0.08 are acceptable.
    2. “After examining the fit of the model for changes in self-esteem among AAWID, the relationship between the change models and demographic variables was verified”. Rather than the usage of “verified”, the model fit should be further explained through the level of fit each statistics represents.
    3. More description on the use of wave 3 in these analyses are needed.

Results:

  1. Based on comments above, it is unclear what data is being used for the initial baseline data of self-esteem and wave 3 should be further detailed in the methods section.
  2. Rather than say the “suitability of the model was verified”, I would suggest further clarification of the model fit, such as “adequate, acceptable, good, etc.”
  3. Use consistent terminology when describing results. For example, “Wave 2” and “second wave” should be consistent if reported the same results.
  4. Table 2
    1. Table 3 is cited in the text before table 2.
    2. TLI is reported in the results section and in the notes section of table 2 but not reported in the table itself, and not described in the methods section.
    3. The RMSEA for the linear model is .155 but it is not included in the provided range (1.00- .218). Please double check these statistics.
  5. Table 3:
    1. Table 3 heading is not left aligned for consistency with other tables.
    2. The statistics reported under variation model is not aligned with a column .
    3. Be consistent with the use of capitalization throughout the variables in your tables.
  6. Table 4: There are two unstandardized columns under slope.
  7. Figure 1. The intercept, and slope appears to be cut off or incomplete in this figure.
    1. It is unclear what the “0”s and “1”’s represent in this figure.

Discussion:

  1. The presentation of the summary of results, and comparisons to previous work and inconsistent results are thoroughly reported.
  2. The non-linear change presented is interesting, and important but it is hard to connect back to your results given the inconsistency in methods used. More clarity and flow of methodology to results to discussion is needed to better understand what time points were used to examine rates of change.
  3. Given the reported findings on school-age children in the discussion, what was the reasoning for using age 15 as the starting age point in this study?
  4. The limitation regarding missing demographic variables is important. The authors should elaborate on some of the key variables that need to be studied in future work here.
    1. I am curious on the impact of their disability on self-esteem. Are there existing variables or measures that can be examined to better understand how their disability may impact their self-esteem?
  5. An implications section that includes the importance of self-esteem and the potential impact of measuring and understanding self-esteem in individuals with ID over time seems to be missing from the end of this manuscript.

Author Response

Overall: Brief Synopsis: This manuscript presents findings from a longitudinal analysis examining the change of self-reported levels of self-esteem in a sample of individuals with intellectual disabilities. This study uses data from a Panel Survey through the Korea Employment Agency for the Disabled. Findings suggest self-esteem in adolescents and adults with intellectual disabilities demonstrate a non-linear change over time, and females reported lower rates of self-esteem yet faster rates of changes compare to males with intellectual disabilities. The discussion focuses on the comparison of findings to previous, inconsistent results, and suggests future direction needed based on the identified limitations of the current study. 

Response: I sincerely thank you for taking the time to carefully review this manuscript. I read your valuable opinions, carefully reviewed the contents of the manuscript accordingly, and revised them in good faith. I would like to say that your advice has been very helpful in improving the completeness of this study, and I would like to express my sincere gratitude once again. The revised text in the manuscript has been highlighted in blue font. The responses to all comments have been prepared and attached herewith/given below.

Comment #1: Abstract: Must be updated and clarified based on the comments below. Make sure to spell out AAWID before using the abbreviation (avoid using abbreviation in the abstract).

Response #1: I have checked and revised it (Throughout manuscript).

Comment #2: Introduction: The introduction does a good job on explaining the differential definitions of self-esteem and theories of the trajectory of self-esteem across the lifespan. Consider expanding the introduction to address specific findings for individuals with ID or other related factors (such as self-determination, quality of life, etc.). It appears there is a lack of knowledge of the trajectory of self-esteem in individuals with ID but it may be helpful to briefly describe the current state of what is known about self-esteem generally in ID and its association with larger constructs such as well-being or quality of life.

Response #2: In reflection of the opinions, the introduction to the discussion was revised throughout and content on related literature was added (page 1, line 53 ~ page 2, line 2; page 2, line 25 ~ line 46; page 2, line 48 ~ page 3, line 31).

Comment #3: Methods:                                                             

Comment #3-1: There is some inconsistency in the design and data used in this study. In the “Study Design” subsection of the Materials and Methods section, The author states the study employed “a cross-sectional survey design”. In the “Data” subsection, the authors state they used data from wave 2 of the survey. However, based on the details provided further into the methods and the results, it does not appear this study is a cross-sectional survey design, and waves 1 and 3 were also reported. More detail is needed to describe each wave of data used, the time in between data collection, and any attrition between the data collection periods.

Response #3-1: Thanks for the correct correction. I modified it to longitudinal study and changed it to measurement to avoid confusion caused by the term wave. I added information about each data collection period and also provided information about subject dropout (page 3, line 42; page 5, line 20 ~ line 31; Throughout manuscript).

Comment #3-2: What is the rationale for the age ranges and age groupings selected?

Response #3-2: The rationale for selecting the age ranges has been added to the introduction (page 3, line 12 ~ line 23).

Comment #3-3: Self-Esteem Scale: Given the challenges they may arise from individuals with intellectual disabilities ability to self-report, were there any tests of comprehension or adaptations made to ensure the individuals were able to effectively and accurately report on their own experiences? It may be helpful to report psychometrics related to individuals with intellectual disabilities for this scale? It appears this scale originally uses a 4-point Likert scale, was this an adaptation made by the author?

Response #3-3: The measurement data of the self-esteem scale in the first year have been confirmed to have metrological suitability in previous studies. A detailed description of this has been added. We appreciate the correction of errors. The self-esteem scale is a 4-point scale, and the description of this has been revised (page 4, line 41 ~ page 5, line 6).

Comment #3-4: The severity level of Intellectual disability would be beneficial to report with this sample and determine if level of ID impacts in changes of self-esteem over time.

Response #3-4: Information on the degree of intellectual disability is presented by inserting it into the table. Although information on disability grade is provided in the panel data, the disability grade system for intellectual disability has been abolished in South Korea and is divided into severe and mild. All individuals with intellectual disabilities are severe (Table 1, Figure 1).

Comment #3-5: Statistical Analysis:

“the comparative fit index (CFI), normed fit index (NFI), and root mean square error of approximation (RMSEA) were used as fitness indices. If the fit indices are >.90, the model is considered to be a good fit. In the case of RMSEA, a good fit is when the value <.10 is considered a bad fit”. These sentences are unclear and seem to be contradictory. RMSEA’s values > .10 indicates a poor fit, while 0.05 and 0.08 are acceptable.

“After examining the fit of the model for changes in self-esteem among AAWID, the relationship between the change models and demographic variables was verified”. Rather than the usage of “verified”, the model fit should be further explained through the level of fit each statistics represents. More description on the use of wave 3 in these analyses are needed.

Response #3-5: Thank you for the correction. I have revised it according to your comment, and replaced it with the term measurement due to the confusion of the term wave (page 5, line 20 ~ line 31; page 5, line 39 ~ line 42; Throughout manuscript).

Comment #4: Results:

Comment #4-1: Based on comments above, it is unclear what data is being used for the initial baseline data of self-esteem and wave 3 should be further detailed in the methods section.

Response #4-1: Added corrections to the methods section based on good your comments (page 4, line 1 ~ line 6).

Comment #4-2: Rather than say the “suitability of the model was verified”, I would suggest further clarification of the model fit, such as “adequate, acceptable, good, etc.”

Response #4-2: I have revised the terminology overall based on your valuable suggestions (Throughout manuscript).

Comment #4-3: Use consistent terminology when describing results. For example, “Wave 2” and “second wave” should be consistent if reported the same results.

Response #4-3: Due to the confusion surrounding the term wave, the term measurement was used instead and explained (Throughout manuscript).

Comment #4-4: Table 2, Table 3 is cited in the text before table 2; TLI is reported in the results section and in the notes section of table 2 but not reported in the table itself, and not described in the methods section; The RMSEA for the linear model is .155 but it is not included in the provided range (1.00- .218). Please double check these statistics.

Response #4-4: Thanks for pointing out the errors. They have all been corrected (Table 2).

Comment #4-5: Table 3, Table 3 heading is not left aligned for consistency with other tables; The statistics reported under variation model is not aligned with a column; Be consistent with the use of capitalization throughout the variables in your tables.

Response #4-5: Thanks for pointing out the errors. They have all been corrected (Table 3).

Comment #4-6: Table 4, There are two unstandardized columns under slope, Figure 1. The intercept, and slope appears to be cut off or incomplete in this figure, It is unclear what the “0”s and “1”’s represent in this figure.

Response #4-6: Table 4 has been revised and detailed descriptions of the models and constants have been added to the Statistical Analysis section (Table 4; page 5, line 20 ~ line 31).

Comment #5: Discussion:

Comment #5-1: The presentation of the summary of results, and comparisons to previous work and inconsistent results are thoroughly reported.

Response #5-1: Thank you for your kind comment.

Comment #5-2: The non-linear change presented is interesting, and important but it is hard to connect back to your results given the inconsistency in methods used. More clarity and flow of methodology to results to discussion is needed to better understand what time points were used to examine rates of change.

Response #5-2: Added detailed explanation in the statistical analysis section for nonlinear models (page 5, line 20 ~ line 37).

Comment #5-3: Given the reported findings on school-age children in the discussion, what was the reasoning for using age 15 as the starting age point in this study?

Response #5-3: The reason is explained in the introduction and added to the discussion (page 3, line 12 ~ line 23; age 9, line 3 ~ line 11).

.

Comment #5-4: The limitation regarding missing demographic variables is important. The authors should elaborate on some of the key variables that need to be studied in future work here. I am curious on the impact of their disability on self-esteem. Are there existing variables or measures that can be examined to better understand how their disability may impact their self-esteem?

Response #5-4: Additional analysis of the degree of disability was conducted and reflected throughout the manuscript (Table 1; Figure 1; Throughout manuscript).

Comment #5-5: An implications section that includes the importance of self-esteem and the potential impact of measuring and understanding self-esteem in individuals with ID over time seems to be missing from the end of this manuscript.

Response #5-5: An implications section has been added (page 10, line 41 ~ page 11, line 17)

Reviewer 3 Report

Comments and Suggestions for Authors

Thank you for the opportunity to review this manuscript. The current msc is about a topic of relevance and general interest to the readers of the journal (and the world). Your manuscript has some positive features. But, as detailed in the enclosed reviews, there are a number of problems as well and were identified a number of issues with the manuscript, expressing significant concern about both the focus and analysis.

Acronyms: our suggestion is to eliminate some of the acronyms (even if they are accepted by the journal). The space that is saved is not worth the time required to look up each acronym (e.g.: see AAWID). In my opinion, it will be much more understandable.

Introduction: Although introduction is concise, greater attention should be given to the background and theory supporting this research: strengthening the importance of such study based on a more robust literature review about self-esteem (conceptual model). If there is already evidence of self-esteem tend to stabilize or increase during development ages (as stated in page 1) why author consider important to analyze it with persons with ID?  This group definition was not provided. Relevant gaps (what are the limitations of previous research about the topic) should be identified – and not only mentioned that there is lack of evidences. How does this study address these limitations? How does extend previous research? What is its main-value/contribution to research and practice?               

Methodology

Sample:  Please justify the sample size adequacy. Why the second wave? How was persons with ID identified – by having a clinical diagnosis? IQ and adaptive behavior were measured? It would be helpful to know how individuals were differentiated with regards to falling in either the borderline, mild or moderate. Does author have this data? More information on participant characteristics is necessary (where did they lived - institutions, own home..), etc..  Instrument: Although the good description of the assessment used, and understanding the tension between space and content, author is recommended to advance, even if briefly, the validity of the assessments used? it would be good to provide some brief information on (a) psychometric properties and b) systems of collecting data by the original authors. Did persons with ID able to read, understand and answer to the questionnaire? How was the survey administration: interview, self-administered…?

Discussion: should be restructured. What is the rationale for testing 2nd and 3rd wave? And why 2nd wave was chosen?  Why those 3 variables were chosen? Author is recommended to deepen in the explanations of some results, because in most of discussion section author limits to present other studies’ findings. How changes vs. stabilization could be explained? Was there any specific ages characterized by change or by stabilization? Author states that there are individual differences in the rate of change. Maybe these differences could be presented and discussed. Why was sex the variable that influence self-esteem? Was this finding expected? How can be explained? again: data provide evidence of lower self-esteem levels in women with IDD although it change faster than among men. Could author advance with possible explanations for that? At same point a study comparing persons with and without ID is presented but with no clear association with the topic. Why do author consider that self-esteem changes are meaningful  for persons with ID. Limitations should be reinforced. Reccomendations to practice and research should be added.

Conclusions: I would have expected to see more emphasis on the broad contribution and importance of this study. It is suggested that author should sum up the main conclusions found, based on findings, avoiding generalizations that are not possible within the study. What are the practical implications of this study? This section would be greatly enhanced by adding a recommendations sub-section. 

References: the msc only have 3 “recent” references; author is invited to update this section

Author Response

Reviewer 3

Overall: Thank you for the opportunity to review this manuscript. The current msc is about a topic of relevance and general interest to the readers of the journal (and the world). Your manuscript has some positive features. But, as detailed in the enclosed reviews, there are a number of problems as well and were identified a number of issues with the manuscript, expressing significant concern about both the focus and analysis.

Response: I sincerely thank you for taking the time to carefully review this manuscript. I read your valuable opinions, carefully reviewed the contents of the manuscript accordingly, and revised them in good faith. I would like to say that your advice has been very helpful in improving the completeness of this study, and I would like to express my sincere gratitude once again. The revised text in the manuscript has been highlighted in blue font. The responses to all comments have been prepared and attached herewith/given below.

Comment #1: Acronyms: our suggestion is to eliminate some of the acronyms (even if they are accepted by the journal). The space that is saved is not worth the time required to look up each acronym (e.g.: see AAWID). In my opinion, it will be much more understandable.

Response #1: I have minimized the acronyms according to the comments and especially revised all of them related to AAWID.

Comment #2: Introduction: Although introduction is concise, greater attention should be given to the background and theory supporting this research: strengthening the importance of such study based on a more robust literature review about self-esteem (conceptual model). If there is already evidence of self-esteem tend to stabilize or increase during development ages (as stated in page 1) why author consider important to analyze it with persons with ID?  This group definition was not provided. Relevant gaps (what are the limitations of previous research about the topic) should be identified – and not only mentioned that there is lack of evidences. How does this study address these limitations? How does extend previous research? What is its main-value/contribution to research and practice?    

Response #3: The introduction section has been revised to reflect all valuable comments (page 1, line 38 ~ line 51; page 1, line 53 ~ page 2, line 2; page 2, line 7 ~ line 14; page 2, line 25 ~ line 46; page 2, line 47 ~ page 3, line 31).

Comment #3: Methodology Sample:  Please justify the sample size adequacy. Why the second wave? How was persons with ID identified – by having a clinical diagnosis? IQ and adaptive behavior were measured? It would be helpful to know how individuals were differentiated with regards to falling in either the borderline, mild or moderate. Does author have this data? More information on participant characteristics is necessary (where did they lived - institutions, own home..), etc..  Instrument: Although the good description of the assessment used, and understanding the tension between space and content, author is recommended to advance, even if briefly, the validity of the assessments used? it would be good to provide some brief information on (a) psychometric properties and b) systems of collecting data by the original authors. Did persons with ID able to read, understand and answer to the questionnaire? How was the survey administration: interview, self-administered…?

Response #4: A statement on sample adequacy was inserted. The reason for using the second wave was added. The content that the subjects were diagnosed intellectually disabled people was inserted. Both intellectual function and adaptive behavior were considered at the time of diagnosis. Information on the degree of disability was added and additional analysis was conducted. The subjects were intellectually disabled people living in the community, and there were no intellectually disabled people living in facilities. Previous studies on the psychometric properties of the measurement tool were reviewed and explained. The survey method and data review method were added (page 4, line 2 ~ line 6; page 4, line 8 ~ line 11; page 4, line 16 ~ line 35; page 5, line 20 ~ line 31; page; page 4, line 44 ~ page 5, line 6; Table 1; Figure 1).

Comment #4: Discussion: should be restructured. What is the rationale for testing 2nd and 3rd wave? And why 2nd wave was chosen?  Why those 3 variables were chosen? Author is recommended to deepen in the explanations of some results, because in most of discussion section author limits to present other studies’ findings. How changes vs. stabilization could be explained? Was there any specific ages characterized by change or by stabilization? Author states that there are individual differences in the rate of change. Maybe these differences could be presented and discussed. Why was sex the variable that influence self-esteem? Was this finding expected? How can be explained? again: data provide evidence of lower self-esteem levels in women with IDD although it change faster than among men. Could author advance with possible explanations for that? At same point a study comparing persons with and without ID is presented but with no clear association with the topic. Why do author consider that self-esteem changes are meaningful for persons with ID. Limitations should be reinforced. Reccomendations to practice and research should be added.

Response #4: The term of wave has been revised into measurement. The rationale of the choice of the 2nd wave has been added in the method section. The reason for the selection of variables was explained. The 2nd measurement change model was adopted, and gender and disability level were identified as factors affecting the change, and the explanation was expanded. In the discussion, parts that were not related to the results of this study were deleted and organized. The limitation and implication sections were added (page 8, line 21 ~ page 9, line 3; page 9, line 4 ~ line 11; page 9, line 36 ~ line 39; page 10, line 10 ~ line 17; page 10, line 24 ~ page 11, line 17).

Comment #5: Conclusions: I would have expected to see more emphasis on the broad contribution and importance of this study. It is suggested that author should sum up the main conclusions found, based on findings, avoiding generalizations that are not possible within the study. What are the practical implications of this study? This section would be greatly enhanced by adding a recommendations sub-section. 

Response #6: Revised according to your comments. The limitation and implication sections were added (page 11, line 18 ~ line 30; page 10, line 24 ~ page 11, line 17).

Comment #6: References: the msc only have 3 “recent” references; author is invited to update this section

Response #6: Related recent references have been added.

Alkhasawneh, T, Al-Shaar, AS, Khasawneh, M, Darawsheh, S, & Aburaya, N. (2022). Self-Esteem and its Relationship to some Demographic Variables among Students with Learning Disabilities. Information Sciences Letters, 11(6), 1929-1936.

Davies, Lauren, Randle‐Phillips, Cathy, Russell, Ailsa, & Delaney, Claire. (2021). The relationship between adverse interpersonal experiences and self‐esteem in people with intellectual disabilities: The role of shame, self‐compassion and social support. Journal of Applied Research in Intellectual Disabilities, 34(4), 1037-1047.

Schalock, RL, Luckasson, Ruth, & Tassé, MJ. (2021). Twenty questions and answers regarding the 12th edition of the AAIDD manual: Intellectual disability: definition, diagnosis, classification, and systems of supports. American Association on Intellectual and Developmental Disabilities, 1-5.

Round 2

Reviewer 1 Report

Comments and Suggestions for Authors

I congratulate the author for his/her answer to my comments.

I have a few comments that need to be adressed.

- In the abstract, please be more specific when you talke about rate of change. Is-it a significant increase of self-esteem over the two years ? 

- The review of Kaspal is about physical activity. this is not the right reference. I think that the author is talking about the following reference that was presented in the first version of the manuscript: Maïano, C., et al. (2019). Self-concept research with school-aged youth with intellectual disabilities: A systematic review. Journal of Applied Research in Intellectual Disabilities, 32(2), 238–255.

- Please, do not used the term patients for the participants.

- It is fit indexes and not fitness indices.

- Section 3.2. Please, be more specific. Is-it a significant increase of self-esteem over time ? 

- Section 3.3. Please, explained how gender was coded. is-it 0 = male and 1 = female ? 

- Section 3.3. Please be more specific. What does the gender effect mean ? For example, that the significant increase if self-esteem is significant higher in male relative to female ? 

- Figure 1. The information are redundant with Table 4. However, a grapgic representation of the three hypothesized LGM model will be helpful for the reader that is unfamiliar with unconditional and conditional LGM.

Comments on the Quality of English Language

I suggest a revision of the text.

Author Response

I sincerely appreciate your thoughtful and encouraging review of our manuscript. I will carefully review any minor revisions you have suggested to improve the clarity and completeness of the manuscript.

Comment #1: In the abstract, please be more specific when you talk about rate of change. Is-it a significant increase of self-esteem over the two years? 

Response #1: This has been revised accordingly (page 1, line 29 ~ line 31).

Comment #2: The review of Kaspal is about physical activity. this is not the right reference. I think that the author is talking about the following reference that was presented in the first version of the manuscript: Maïano, C., et al. (2019). Self-concept research with school-aged youth with intellectual disabilities: A systematic review. Journal of Applied Research in Intellectual Disabilities, 32(2), 238–255.

Response #2: Thank you for your insightful comment. This has been revised accordingly (page 2, line 41 ~ line 44).

Comment #3: Please, do not used the term patients for the participants.

Response #3: Thank you for your valuable correction. This has been revised accordingly (page 6, line 8 ~ line 9).

Comment #4: It is fit indexes and not fitness indices.

Response #4: Thank you for your valuable correction. This has been revised accordingly (page 6, line 42; page 7, line 10).

Comment #5: Section 3.2. Please, be more specific. Is-it a significant increase of self-esteem over time? 

Response #5: Thank you for your insightful comment. This has been revised accordingly (page 7, line 29 ~ line 31).

Comment #6: Section 3.3. Please, explained how gender was coded. is-it 0 = male and 1 = female? 

Response #6: Information on coding has been added to the text (page 5, line 11 ~ line 12).

Comment #7: Section 3.3. Please be more specific. What does the gender effect mean? For example, that the significant increase if self-esteem is significant higher in male relative to female? 

Response #7: Thank you for your insightful comment. This has been revised accordingly (page 8, line 7 ~ line 10).

Comment #8: Figure 1. The information are redundant with Table 4. However, a grapgic representation of the three hypothesized LGM model will be helpful for the reader that is unfamiliar with unconditional and conditional LGM.

Response #7: Thank you for your insightful comment. Table 4 has been deleted and figure 1 for three hypothesized LGM model has been added.

Reviewer 2 Report

Comments and Suggestions for Authors

I believe the authors have addressed all concerns appropriately, and the manuscript is significantly improved. I approve it in its current form.

Author Response

Comment: I believe the authors have addressed all concerns appropriately, and the manuscript is significantly improved. I approve it in its current form.

Response : I appreciate your positive comments and acceptance of our paper. I also appreciate that we have been able to improve our manuscript by reflecting all of your comments.

Reviewer 3 Report

Comments and Suggestions for Authors

The author have significantly strengthened their article based on the reviewers comments (I would like to congrat for that). The methods and results are clear and concise. I have no additional suggestions.

Author Response

Comment: The author have significantly strengthened their article based on the reviewers comments (I would like to congrat for that). The methods and results are clear and concise. I have no additional suggestions.

Response : I appreciate your positive comments and acceptance of our paper. I also appreciate that we have been able to improve our manuscript by reflecting all of your comments.